# 1 Vulnerability of soil organic matter of anthropogenically disturbed

# 2 organic soils

Annelie Säurich<sup>1</sup>, Bärbel Tiemeyer<sup>1</sup>, Axel Don<sup>1</sup>, Michel Bechtold<sup>1,2</sup>, Wulf Amelung<sup>3</sup>, Annette
 Freibauer<sup>1,4</sup>

5

- <sup>6</sup> <sup>1</sup>Thünen Institute of Climate-Smart Agriculture, Bundesallee 50, 38116 Braunschweig, Germany
- <sup>7</sup> <sup>2</sup>now at: KU Leuven, Department of Earth and Environmental Sciences, Division Soil and Water Management,
   8 Celestijnenlaan 200 E, B-3001 Heverlee, Belgium
- <sup>3</sup>University of Bonn, Institute of Crop Science and Resource Conservation, Soil Science and Soil Ecology, Nussallee 13,
   53115 Bonn, Germany
- <sup>4</sup> now at: Bavarian State Research Center for Agriculture, Institute of Organic Farming, Agricultural Sciences and Natural
- Resources, Lange Point 12, 85354 Freising, Germany
- *Correspondence to*: Bärbel Tiemeyer (baerbel.tiemeyer@thuenen.de)

# 16 Abstract

Drained peatlands are hotspots of carbon dioxide  $(CO_2)$  emissions from agriculture. As a consequence of both drainageinduced mineralisation and anthropogenic mixing with mineral soils, large areas of former peatlands under agricultural use 18 19 now contain soil organic carbon (SOC) at the boundary between mineral and organic soils and/or underwent a secondary 20 transformation of the peat (e.g. formation of aggregates). However, low carbon organic soils have rarely been studied since 21 previous research has mainly focused on either mineral soils or true peat soils. The aim of the present study was to evaluate 22 the soil organic matter (SOM) vulnerability of the whole range of organic soils including very carbon rich mineral soils (73 g 23  $kg^{-1} < SOC < 569 g kg^{-1}$ ) and to identify indicators for mineralisation of such anthropogenically disturbed organic soils. 24 Using a large sample pool from the German Agricultural Soil Inventory, 91 soil samples were selected covering a broad 25 range of soil and site characteristics. Fen and bog samples were grouped into disturbance classes according to their pedogenetic features. Potential CO<sub>2</sub> production by aerobic incubation was then measured. Specific basal respiration rates 26 (SBR) per unit SOC showed the highest potential emissions for heavily disturbed fen  $(12.1 \pm 5.0 \ \mu g \ CO_2-C \ g \ SOC^{-1} \ h^{-1})$  and 27 moderately disturbed bog samples (10.3  $\pm$  5.2  $\mu$ g CO<sub>2</sub>-C g SOC<sup>-1</sup> h<sup>-1</sup>). Surprisingly, SOM vulnerability increased with an 28 29 increasing degree of disturbance and a decreasing SOC content, indicating positive feedback mechanisms as soon as peat soils are disturbed by drainage. Furthermore, with increasing degree of disturbance the variability of the SBR increased 30 31 drastically, but correlations between soil properties and SBR could not be identified. Respiration rates increased more 32 strongly with an increasing degree of disturbance in bog than in fen samples. Peat properties that positively influenced the turnover of SOM in less disturbed soil samples were mainly pH value and nitrogen content, while phosphorus was important 33 34 for the mineralisation of increasingly disturbed samples and bog peat in general. Furthermore, a narrow carbon-to-nitrogen ratio correlated strongly with potential emissions. Given the high potential of  $CO_2$  emissions from organic soils with a low 35 SOC content, mixing with mineral soil does not seem to be a promising option for decreasing emissions. 36

#### 37 1 Introduction

Organic soils worldwide cover approximately 330 million ha or 2.2 % of the global terrestrial surface. One third of this area 39 is in Europe (Tubiello et al., 2016), corresponding to 3 % of the European landmass (Montanarella et al., 2006). Despite the 40 small extent of this area, peatlands store more than one third of global soil organic carbon (SOC) (Gorham, 1991). Moreover, 41 intact peatlands under waterlogged conditions are ongoing carbon (C) sinks in that their mineralisation rates are lower than 42 their biomass production rates (Clymo et al., 1998). Large areas of peatland are drained for agriculture, forestry and peat 43 mining for energy and horticulture. To date 25.5 million ha of peatlands worldwide have been drained for agriculture alone, of which around 60 % are in the boreal or temperate zone (Tubiello et al., 2016). The majority of the drained peatlands in 44 45 Russia, Belarus, Ukraine, Poland, the Netherlands and Germany are used for agricultural purposes, primarily as grassland (Joosten and Clarke, 2002). These anthropogenic impacts lead to a disturbance of the peatlands' hydrological and 46 biogeochemical cycles, e.g. to the destabilisation of the soil organic matter (SOM) (Holden et al., 2004). Thus drainage turns 47 48 peatlands into net greenhouse gas (GHG) sources, which emit large amounts of carbon dioxide (CO<sub>2</sub>) and nitrous oxide 49  $(N_2O)$  (Maljanen et al., 2010; Tiemeyer et al., 2016).

Furthermore, drainage alters physical and chemical peat properties considerably. The loss of buoyancy following drainage 51 leads to compaction and thus to an increase in bulk density and decrease in total porosity (Rovdan et al., 2002). Compaction 52 and mineralisation jointly cause subsidence of the soil surface. Mineralisation and transformation of SOM lead to the 53 formation of aggregates, shrinkage cracks, earthification and finally to the formation of a dusty, fine-grained ("moorshy") 54 horizon (e.g. Ilnicki and Zeitz, 2003). Consequently, the majority of topsoils of drained agricultural peatlands show a von 55 Post decomposition degree of H10 (von Post, 1924).

Drainage favours carbon over nitrogen (N) mineralisation and microbial N immobilisation during decomposition. Thus, the 57 concentrations of N increase, and both the C concentration and C:N-ratio decrease with increasing degrees of SOM 58 decomposition, especially in the topsoil (Wells and Williams, 1996). Phosphorus (P) concentrations usually increase after 59 drainage, while potassium (K), calcium (Ca) and iron (Fe) concentrations decrease (Sundström et al., 2000; Wells and 50 Williams, 1996; Zak et al., 2010). As aerobic decomposers preferably use the lighter <sup>12</sup>C for respiration, the remaining peat 51 is enriched in <sup>13</sup>C (Ågren et al., 1996). Similarly, drained peatlands are depleted in <sup>14</sup>N and show increases in  $\delta^{15}$ N values

- (Krüger et al., 2015).
- Besides drainage, the conversion from pristine peatlands to agricultural land can comprise the active enrichment of the mineral soil fraction in the top peat layer in order to enhance trafficability. This can be achieved by mixing with mineral soil layers underlying the peat (deep ploughing) or by surface application of mineral soil with or without subsequent ploughing (Göttlich, 1990; Okruszko, 1996). As a consequence of both drainage-induced mineralisation and anthropogenic mineral soil
- mixing, especially the topsoils of large areas of former peatlands under agricultural use contain SOC at concentrations
- between those of mineral and organic soils (Schulz and Waldeck, 2015).

As previous investigations mainly have focused either on mineral (< 150 g SOM kg<sup>-1</sup> according to the German definition, 69 Ad-Hoc-AG Boden, 2005) or "true" peat soils (> 300 g SOM kg<sup>-1</sup>), there are very few studies on soil properties or SOM 70 dynamics of "low C organic soils" (between 150 and 300 g SOM kg<sup>-1</sup>). However, measurements of GHG emissions in the 71 field have shown that organic soils with a SOC content of around 100 g kg<sup>-1</sup> still emit large amounts of CO<sub>2</sub> similar to the 72 levels emitted by "true" peat soils (Leiber-Sauheitl et al., 2014; Tiemeyer et al., 2016). This is rather surprising as the 73 74 remaining organic matter should not be readily available for mineralisation, given that the SOC content at this stage of 75 decomposition is fairly low and CO<sub>2</sub> emissions and SOC content are closely related in mineral soils (Don et al., 2013; Wang et al., 2003). However, only a few field studies have been carried out and there is therefore very limited knowledge about the 76 77 separate effects of climate, hydrology and soil properties. The reasons behind the relatively high  $CO_2$  emissions of the whole 78 continuum of organic soils, including those bordering mineral ones, are not yet clear. Peat properties and SOM quality obviously influence the CO<sub>2</sub> emissions of disturbed peatlands (Brouns et al., 2016; Laiho, 2006), but there is a lack of any 79 80 systematic evaluation of the vulnerability of a wide range of organic soils, including strongly disturbed ones.

The aims of this study, were i) to assess the sensitivity of SOM from anthropogenically disturbed organic soils under agricultural use to mineralisation under aerobic conditions and ii) to determine the indicators and drivers of the vulnerability of SOM. In this context, disturbance was defined as the effect of soil-forming processes induced by drainage and/or by the mixing of peat with mineral soil. For this purpose, 91 samples of soils were examined under cropland and grassland from across Germany, ranging from carbon-rich mineral soil (70 g SOC kg<sup>-1</sup>) to "true" peatlands (up to 560 g SOC kg<sup>-1</sup>). As a simulation of the potential effects of drainage, all the samples were aerobically incubated in the laboratory at standardised water content.

# 89 2 Material and Methods

#### 90 2.1 Sample selection

The samples used in this study came from the German Agricultural Soil Inventory, the aim of which is to improve 92 understanding of SOC stocks in agricultural soils. During the soil inventory, agricultural soils in Germany were sampled 93 following standardised protocols in an 8x8 km grid (> 3,000 sites) at seven depth increments per soil pit (10, 30, 50, 70, 100, 94 150 and 200 cm). All the samples were analysed for SOC and bulk density, as well as for basic explanatory soil properties (Table 1, see section 2.2). 91 samples of organic soil horizons from 67 sites were selected. The basic criteria were a SOC 95 96 content > 70 g kg<sup>-1</sup> and a sampling depth > 10 cm to reduce the influence of potential root biomass residues in the samples. 97 Roots have generally been separated by hand, but this might be challenging in organic soils. The final sample selection was 98 based on a cluster analysis to optimally cover the total parameter range, as well as land use, peat type (bog/fen) and geographical position. The selected samples included croplands (19%) and grasslands (81%), which correspond to the 99 dominant agricultural land use of organic soils in Germany. In addition to the 91 samples, ten samples of three 100 anthropogenically undisturbed peatlands (bog, transition bog and fen) were sampled. 101

**Table 1:** Soil properties of the selected soil samples as medians and standard errors. Standard parameters measured in the German 103 Agricultural Soil Inventory: SOC: soil organic carbon content, N<sub>t</sub>: total nitrogen content, CaCO<sub>3</sub>: calcium carbonate content, C:N-ratio: 104 carbon to nitrogen ratio,  $\rho$ : bulk density, pH-value, texture (\* only determined for samples with SOC < 172 g kg<sup>-1</sup>). Additional parameters 105 of this study: Fe<sub>0</sub>: oxalate extractable iron oxide content, P<sub>CAL</sub>: calcium acetate lactate (CAL) extractable phosphorus content,  $\delta^{13}$ C and 106  $\delta^{15}$ N.

| Parameter                                                       | Median                     | Min.   | Max.   |  |
|-----------------------------------------------------------------|----------------------------|--------|--------|--|
| SOC (g kg <sup>-1</sup> )                                       | 257.0 ± 15.5 73.4          |        | 568.9  |  |
| $N_t (g kg^{-1})$                                               | $11.7\pm0.8$               | 2.9    | 36.5   |  |
| $CaCO_3 (g kg^{-1})$                                            | $0.0 \pm 1.2$              | 0.0    | 580.0  |  |
| C:N-ratio                                                       | $18.0\pm1.2$               | 9.9    | 72.6   |  |
| ρ (g cm <sup>-3</sup> )                                         | $0.30\pm0.03$              | 0.07   | 0.99   |  |
| pH CaCl <sub>2</sub>                                            | $4.9\pm0.1$                | 2.5    | 7.4    |  |
| Sand content (%)*                                               | $44.7\pm4.9$               | 2.5    | 87.9   |  |
| Silt content (%)*                                               | $25.8\pm2.5$               | 6.4    | 62.1   |  |
| Clay content (%)*                                               | $20.9\pm3.1$               | 3.9    | 62.8   |  |
| $\operatorname{Fe}_{O}(\operatorname{g}\operatorname{kg}^{-1})$ | $11.8 \pm 1.7$             | 0.4    | 108.3  |  |
| $P_{CAL} (mg kg^{-1})$                                          | $12.4\pm5.9$               | 0.4    | 365.6  |  |
| δ <sup>13</sup> C (‰)                                           | $\textbf{-28.14} \pm 0.10$ | -30.42 | -25.47 |  |
| $\delta^{15}$ N (‰)                                             | $2.05\pm0.24$              | -2.55  | 11.23  |  |

# 107 2.2 Soil properties

- Concentrations of total C and N (N<sub>t</sub>), as well as the total inorganic carbon content for samples with carbonate ( $pH_{CaCl2} > 6.2$ ), 109 were measured by dry combustion (RC 612, LECO Corporation, St. Joseph, USA).
- Stable isotope analysis ( $\delta^{13}$ C and  $\delta^{15}$ N) was performed using a mass spectrometer coupled with an elemental analyser
- (Isoprime 100 and Vario Isotope, Elementar, Hanau, Germany) via a continuous flow system. Samples containing carbonates
- underwent a carbonate destruction (volatilisation method) on the basis of Hedges and Stern (1984) and Harris et al. (2001)
- prior to isotope analysis. The isotope ratios are expressed in per mill,  $\delta^{13}$ C relative to VPDB standard and  $\delta^{15}$ N relative to
- atmospheric nitrogen standard.
- Poorly crystalline and organically-bound iron oxides (Fe<sub>0</sub>) were extracted with an acidic ammonium oxalate solution. The
- extraction took place in the dark to avoid photo-reduction of ferrous iron oxides (Schwertmann, 1964). The concentration of
- extracted Fe<sub>o</sub> was measured by atomic absorption spectrometry (AA-280FS, Varian, Palo Alto, USA).
- Plant-available concentrations of phosphorus were determined by calcium acetate lactate (CAL) extraction (Schüller, 1969).
- P<sub>CAL</sub> concentrations were measured using the molybdenum blue method (Murphy and Riley 1962).
- The fractions of the texture classes sand, silt and clay were quantified by a semi-automated sieve-pipette machine (Sedimat
- 4-12, UGT, Müncheberg, Germany) after aggregate destruction and the removal of salt and SOM using H<sub>2</sub>O<sub>2</sub> (Vos et al.,
- 2016). Undisturbed soil samples in rings were dried at 105 °C until constant mass and subsequently weighed to determine
- bulk density ( $\rho$ ). The pH values were measured using 0.01 mol/L CaCl<sub>2</sub> and a glass electrode. Electrical conductivity (EC)
- was determined in a water solution.

# 125 2.3 Incubation experiments: Basal respiration and substrate-induced respiration

The soil samples were incubated aerobically under optimum moisture and constant temperature (23 °C) conditions to

- determine basal soil respiration (BR) and substrate-induced respiration (SIR), which was used to calculate microbial biomass
- (Anderson and Domsch, 1978).
- The dried (40 °C) and 2 mm sieved soil samples were moistened to a standardised water content of 60 % water-filled pore 130 space. The apparent porosity of the dried and sieved sample was determined from the bulk density of the loose sample. The 131 necessary amount of water to reach 60 % water-filled pore space was then applied to the soil samples under continuous stirring to ensure uniform rewetting. Afterwards, the moistened samples were stored in darkness under aerobic conditions for 132 7 days at 6 °C and then for a further 7 days at 23 °C for pre-incubation. On day 14, the soil samples were prepared for 133 134 incubation in a semi-automatic incubation device using its flow-through mode (Heinemeyer et al., 1989). Three replicates 135 (20 g dry wt.) of each sample were put loosely in acrylic glass tubes (4 cm diameter) and enclosed at both ends with polystyrene foam stoppers. Humidified ambient air flowed through 24 independent lines containing the soil samples at flow 136
- rates between 160 and 180 ml min<sup>-1</sup>. An infrared CO<sub>2</sub> gas analyser (ADC-255-MK3, Analytical Development Co. Ltd.,

- Hoddesdon, UK) was used to measure CO<sub>2</sub> concentrations. Each replicate sample was measured hourly over an incubation
- time of at least 40 h or until a relatively constant BR was reached (up to 90 h).
- Afterwards soil samples were amended with a mixture of 100 mg glucose and 100 mg talcum using an electronic stir for 30 s
- to determine the active microbial biomass using the SIR method. The mixture was then incubated again for 6 h to obtain the
- maximal initial respiratory response of the microbial biomass (Anderson et al., 1995).

# 143 2.4 Data analysis

Statistical analysis was performed using the R software environment (version R-3.1.3, R Core Team, 2015).

# 145 **2.4.1 Determination of basal soil respiration**

The measured BR is expressed as  $\mu g \operatorname{CO}_2$ -C g soil<sup>-1</sup> h<sup>-1</sup> and the specific basal respiration (SBR) is normalised by the 147 sample's SOC content into  $\mu g \operatorname{CO}_2$ -C g SOC<sup>-1</sup> h<sup>-1</sup>. An exponential model was fitted simultaneously to all three incubation

- replicates to determine the equilibrium values of the SBR (Figure):
- $CO_2 C(t) = a (a SBR) (1 e^{-k * t}),$  (1)
- where CO<sub>2</sub>-C (t)  $[\mu g CO_2$ -C g SOC<sup>-1</sup> h<sup>-1</sup>] is the specific CO<sub>2</sub> production per hour,  $a [\mu g CO_2$ -C g SOC<sup>-1</sup> h<sup>-1</sup>] is the initial
- respiration and k [h<sup>-1</sup>] is the change rate of SBR.
- To achieve an objective quantification of the (specific) basal respiration and its uncertainty, the R package "dream" was used 153 (Guillaume and Andrews, 2012), which is based on the iterative Markov Chain Monte Carlo (MCMC) approach. This 154 method is basically a Markov chain that generates a random walk through the high-probability-density region in the parameter space, separating behavioural from non-behavioural solutions following the probability distribution (Vrugt et al., 155 156 2009b). The differential evolution adaptive metropolis (DREAM) algorithm is an efficient MCMC sampler that runs 157 multiple Markov chains simultaneously for global exploration of the parameter space. In doing so, DREAM uses a differential algorithm for population evolution and a metropolis selection rule to decide whether a population of candidate 158 159 points is accepted or not. After the burn-in period, the convergence of individual chains is checked using the Gelman and Rubin (1992) convergence criterion, which examines the variance between and within chains (Vrugt et al., 2008, 2009a). 160
- Once the convergence criterion of Gelman and Rubin was < 1.01, another 500,000 simulations were run to determine the
- posterior probability density functions of the model parameters, which were used to calculate the median and the 2.5 and 163 97.5 % quantiles.
- For the evaluation of the SIR experiment, the value of the maximum initial respiratory response was identified manually and
- then transcribed to microbial biomass (SIR- $C_{mic}$ ) [µg g<sup>-1</sup> soil] as follows (Kaiser et al., 1992):
- SIR- $C_{mic} = \mu l CO_2 g^{-1} soil * 30.$

- (2)
- To quantify the efficiency of microbial respiration per unit biomass, the metabolic or respiratory quotient  $q(CO_2)$  [mg CO<sub>2</sub>-168 C h<sup>-1</sup> g<sup>-1</sup> biomass SIR-C<sub>mic</sub>] was calculated by dividing the BR by the SIR-C<sub>mic</sub> (Anderson and Domsch, 1985):

(3)

The higher the value of  $q(CO_2)$ , the higher the  $CO_2$  emissions per unit microbial biomass, indicating a lack of available C for 171 metabolism in the soil.

# 172 2.4.2 Degree of disturbance

 $q(CO_2) = \frac{BR}{SIR - C_{mic}/1000}$ 

The present classification of anthropogenic disturbance is based on the mapped soil horizons from which the samples originated. The soil horizons and the degree of decomposition after von Post were mapped according to the German manual 174 175 of soil mapping (Ad-Hoc-AG Boden, 2005). While the original von Post scale was developed for undrained peat, the 176 German classification frequently uses the von Post scale for drained peat as well. The samples of peatland genesis were 177 divided into five different disturbance classes according to the severity of disturbance and (Table 2, based on Ilnicki and Zeitz (2003) and Ad-Hoc-AG Boden (2005)): no disturbance (D0F/D0B), slight disturbance (D1F/D1B), moderate 178 179 disturbance (D2F/D2B), strong disturbance (D3F) and heavy disturbance (D4F). Slightly disturbed horizons experience 180 drainage and are influenced by a fluctuating water table, thus they are temporarily subjected to aerobic conditions but there has not vet been a secondary transformation of the peat structure. Earthified topsoils are defined as "moderately disturbed". 181 182 Strong disturbance is characterised by blocky to prismatic aggregates and/or the formation of shrinkage cracks, and is only found in subsoils. In the present sample set, this level of disturbance only occurred in fen peat. Finally, both highly 183 decomposed dusty "moorsh" and mixtures of peat and mineral soil have been defined as "heavily disturbed". This class also 184 185 only occurred in fen peat. Overall there are five fen classes, three bog classes, and one class each for gyttja (organic or 186 calcareous sediments) and other samples. Samples from the class other were organic marsh soils or could not be assigned to 187 any disturbance class (e.g. buried organic soils). For further information see Table A1 in the appendix. Given that this classification was developed after the sample selection, the distribution among the groups is not uniform. The von Post scale 188 189 from H1 to H10 was altered by adding H11 for low C organic soils deriving from peat. Gyttja and other remaining samples 190 were not included in the van Post scale.

Table 2: Classification of the anthropogenic disturbance and corresponding median and standard error of soil properties: SOC: soil organic carbon content, C:Nratio: carbon to nitrogen ratio,  $\delta^{15}$ N, N<sub>t</sub>: total nitrogen content, P<sub>CAL</sub>: calcium acetate lactate (CAL) extractable phosphorus content, pH-value, EC: electrical conductivity, Fe<sub>0</sub>: oxalate extractable iron content,  $\rho$ : bulk density 192

| Degree of disturbance                                                 | Description                                              | Peatland<br>type | Label | n  | SOC<br>(g kg <sup>-1</sup> ) | C:N        | δ <sup>15</sup> N<br>(‰) | $N_t$<br>(g kg <sup>-1</sup> ) | P <sub>CAL</sub><br>(mg kg <sup>-1</sup> ) | pH            | EC<br>(µS cm <sup>-1</sup> ) | Fe <sub>0</sub><br>(g kg <sup>-1</sup> ) | ρ<br>(g cm <sup>-3</sup> ) |
|-----------------------------------------------------------------------|----------------------------------------------------------|------------------|-------|----|------------------------------|------------|--------------------------|--------------------------------|--------------------------------------------|---------------|------------------------------|------------------------------------------|----------------------------|
| No Pristine or<br>disturbance nearly natura                           | Pristine or                                              | Fen              | D0F   | 12 | $462\pm37$                   | $23\pm3$   | $0.3\pm0.5$              | $19.4\pm2.6$                   | $7.9\pm2.2$                                | $5.5\pm0.4$   | $140\pm38$                   | $5.5\pm1.1$                              | $0.13\pm0.01$              |
|                                                                       | nearly natural                                           | Bog              | D0B   | 9  | $521\pm13$                   | $57\pm8$   | $1.2\pm1.0$              | $9.1\pm2.4$                    | $5.7\pm5.7$                                | $3.2\pm0.1$   | $92\pm23$                    | $0.9 \pm 1.7$                            | $0.10\pm0.01$              |
| Alternatin;<br>Slight aerobic-<br>disturbance anaerobic<br>conditions | Alternating<br>aerobic-                                  | Fen              | D1F   | 9  | $426\pm19$                   | $18\pm4$   | $1.2\pm1.0$              | $21.8\pm2.0$                   | $6.6\pm3.9$                                | $5.2\pm0.4$   | $282\pm124$                  | $13.4\pm2.7$                             | $0.20\pm0.02$              |
|                                                                       | conditions                                               | Bog              | D1B   | 5  | $473\pm13$                   | $46\pm7$   | $0.5\pm0.7$              | $10.7\pm1.5$                   | $41.8\pm13.2$                              | $3.5\pm0.1$   | $70\pm11$                    | $3.3\pm2.8$                              | $0.12\pm0.02$              |
|                                                                       |                                                          | _                |       | _  |                              |            |                          |                                |                                            |               |                              |                                          |                            |
| Moderate<br>disturbance Earthif                                       | Earthification                                           | Fen              | D2F   | 7  | $340 \pm 32$                 | $17 \pm 1$ | $1.2 \pm 0.6$            | 21.0 ± 1.9                     | $12.4 \pm 8.6$                             | $5.2 \pm 0.2$ | $292 \pm 276$                | $21.0 \pm 4.7$                           | $0.30 \pm 0.05$            |
|                                                                       |                                                          | Bog              | D2B   | 6  | $333\pm52$                   | $23\pm2$   | $2.0\pm0.5$              | $14.2\pm2.6$                   | $77.3\pm27.0$                              | $3.8\pm0.3$   | $121\pm10$                   | $9.3\pm1.8$                              | $0.18\pm0.10$              |
| Strong<br>disturbance                                                 | Polyhedral<br>aggregates or<br>cracks                    | Fen              | D3F   | 5  | 320 ± 19                     | $14 \pm 1$ | $1.9\pm0.5$              | $24.0 \pm 1.8$                 | $24.2 \pm 19.9$                            | $5.6\pm0.6$   | 271 ± 55                     | $24.4 \pm 14.1$                          | $0.26\pm0.07$              |
| Heavy<br>disturbance                                                  | Dusty<br>moorsh or<br>high content<br>of mineral<br>soil | Fen              | D4F   | 19 | $142\pm12$                   | $14 \pm 1$ | $3.3 \pm 0.4$            | $10.0 \pm 1.1$                 | $30.9\pm24.5$                              | $5.4\pm0.3$   | $209 \pm 153$                | $19.6\pm2.7$                             | $0.65\pm0.05$              |
| Gyttja                                                                | Organic or<br>calcareous<br>sediments                    | -                | G     | 12 | $100\pm19$                   | $20\pm3$   | $1.7\pm0.6$              | $5.4\pm0.7$                    | $14.2\pm7.5$                               | $4.7\pm0.4$   | $253\pm206$                  | $17.1\pm9.1$                             | $0.51\pm0.08$              |
| Other                                                                 | e.g. organic<br>marsh soils,<br>buried<br>horizons       | -                | 0     | 17 | 124 ± 15                     | $16\pm2$   | $3.5\pm0.5$              | $7.0\pm 0.7$                   | $11.3\pm6.1$                               | $5.2 \pm 0.3$ | 124 ± 146                    | $21.8\pm3.9$                             | $0.54\pm0.05$              |

# 195 2.4.3 Statistical and multivariate analysis

In a first step, Spearman's rank correlation coefficient  $r_s$  was evaluated for the specific basal respiration and all measured explanatory variables using the R package "Hmisc" (Harrell, 2016). The p-values were adjusted using the method after Bonferroni. Differences between the results of disturbance classes for BR, SBR, SIR-C<sub>mic</sub> and q(CO<sub>2</sub>) were determined using an analysis of variance. P-values were computed with the Tukey 'honest significant differences' test ( $\alpha = 0.05$ ) and adjusted with the Bonferroni correction using the R package "multcomp" (Hothorn et al., 2008). Correlation coefficients were classified as follows:  $0.3 \ge r_s \ge 0.7$  is a moderate correlation and  $r_s > 0.7$  a strong correlation.

In a second step, multi-variate linear regression was applied to identify the most crucial factors and interactions influencing 203 the specific basal respiration. The generalised least squares (gls) model was used with the exponential variance structure following the protocol of Zuur et al. (2009) using the R package "nlme" (Pinheiro et al., 2015). The SBR was set as the 204 dependent variable, with the explanatory variables of SOC, N<sub>t</sub>, C:N-ratio, P<sub>CAL</sub>, carbonate (CaCO<sub>3</sub>), pH,  $\delta^{13}$ C,  $\delta^{15}$ N, Fe<sub>0</sub>,  $\rho$ , 205 soil depth and EC as well as the categorical variables of the degree of decomposition (von Post, 1924), the degree of 206 207 disturbance and peatland type. Important meaningful interactions were also incorporated. To achieve homoscedasticity of residuals, the data of EC and P<sub>CAL</sub> were log10 transformed and the C:N-ratio log-transformed. Applying the top-down 208 209 strategy for the model selection, the complete generalised least squares model was run first and variables stepwise removed 210 until all the variables were significant with a p-value

#### 215 **3** Results

# 216 **3.1 Vulnerability of SOM as determined by respiration rates**

For all classes, BR was highly variable, ranging from 0.3 to 7.0  $\mu$ g CO<sub>2</sub>-C g soil<sup>-1</sup> h<sup>-1</sup> (Fig. 1a). The BR rates of fen samples decreased with an increasing degree of disturbance due to concomitantly decreasing SOC content (Table 2), while bog samples behaved inversely. Overall, bog samples (2.0 ± 0.3  $\mu$ g CO<sub>2</sub>-C g soil<sup>-1</sup> h<sup>-1</sup>) had similar BR rates to fen samples (2.5 ± 0.2  $\mu$ g CO<sub>2</sub>-C g soil<sup>-1</sup> h<sup>-1</sup>). *Gyttja* (1.3 ± 0.3  $\mu$ g CO<sub>2</sub>-C g soil<sup>-1</sup> h<sup>-1</sup>) and *other* samples (1.1 ± 0.2  $\mu$ g CO<sub>2</sub>-C g soil<sup>-1</sup> h<sup>-1</sup>) showed significantly lower BR rates than undisturbed, slightly and strongly disturbed fen samples (D0F, D1F, D3F).

Overall, fen samples had significantly higher (p < 0.01) average SBR rates of  $8.3 \pm 0.7 \ \mu g \ CO_2$ -C g SOC<sup>-1</sup> h<sup>-1</sup> than bog 222 samples (5.1  $\pm$  0.9 µg CO<sub>2</sub>-C g SOC<sup>-1</sup> h<sup>-1</sup>). This difference was especially clear for undisturbed and slightly disturbed 223 samples. SBR rates were also highly variable between and within classes, and ranging from 1.5 to 25.1 µg CO<sub>2</sub>-C g SOC<sup>-1</sup> h<sup>-</sup> 224 <sup>1</sup> (Fig. 1b). SBR rates tended to increase with increasing soil disturbance for both fen and bog. D0B samples had significantly 225 lower (p < 0.01) SBR rates than the classes D4F, gyttja and other with  $3.7 \pm 0.6 \ \mu g \ CO_2$ -C g SOC<sup>-1</sup> h<sup>-1</sup>. The moderately 226 227 disturbed D2B samples showed much higher SBR rates  $(10.1 \pm 2.1 \ \mu g \text{ CO}_2\text{-C} \text{ g soil-SOC}^{-1} \text{ h}^{-1})$  than the other two bog classes, and were comparable to strongly and heavily disturbed fen samples (D3F, D4F). Both gyttja and other showed high 228 and variable SBR rates with  $14.2 \pm 2.4 \ \mu g \ \text{CO}_2$ -SOC g SOC<sup>-1</sup> h<sup>-1</sup> and  $10.8 \pm 1.5 \ \mu g \ \text{CO}_2$ -C g SOC<sup>-1</sup> h<sup>-1</sup> respectively. 229

#### 230 **3.2.** Organic matter quality and soil characteristics determining SOM vulnerability

The significant Spearman correlation coefficients indicated moderate positive correlations between SBR rates and phosphorus concentration ( $r_s = 0.52$ ), pH value ( $r_s = 0.42$ ), iron oxides ( $r_s = 0.42$ ) and carbonate concentration ( $r_s = 0.36$ ). Significant negative dependence was found between SBR and the C:N-ratio ( $r_s = -0.62$ ) and SOC content ( $r_s = -0.49$ ) (Fig. 2). The most influencing factors on SBR rates as indicated by the best fitted *gls* model were SOC, C:N-ratio, P<sub>CAL</sub> and  $\rho$  (all p