# Peer review of "Vulnerability of soil organic matter of anthropogenically disturbed"

_Biogeosciences, 2017_

## Referee Comment (RC1) · Anonymous Referee #1 · 31 May 2017

The article deals with decomposition of organic matter in disturbed organic soils. In general the article is too long, authors could consider to make the article concise and focused on key results. In the current form it is too hard to get bottom of the core results found. In addition to this, I have the following issues regarding the article 1. According to the authors, soil samples from Soil Inventory were used for the analysis, but did not mention the year of the sample collection. How old the samples were and was the van Post's disturbance class was identified during sample collection? Or did the authors assign the disturbances classes of the archive soils they used? It seems from Table A1 the disturbance classes need to be identified in field condition. 2. The description of results is confusing; for example "D0B samples had significantly lower (p < 0.01) SBR rates than the classes D4F," was it from the statistical analysis done with GLS model described in statistics section? Or from ANOVA and TukeyHSD?

The experimental design was unbalanced, authors need to describe how they handle "unbalanced" issue. What are the explanatory variables involved in ANOVA. Another example (not limited to) of confusing description of results can be found in L231-236. Were those correlation coefficients significant? Need to mention the step wise variable section using GLS to identify the variables that explained most variation. Authors need to think about making their description and presentation less convoluted for the readers. L249-251 was it based on Spearman's rank correlation? L253-259 need to rewrite in clear concise manner. Figure 2 and 3: was there any difference in data presented? Or same data repeated. 3. It would be better if the authors used the macromolecular quality not only CN ratio of organic matter to assess the effect of disturbance on SOC decomposition. 4. Equation in L149 doesn't make sense 5. what does L143-144 refer to?

---

## Referee Comment (RC2) · Anonymous Referee #2 · 31 May 2017

The authors report on the $CO_2$ release from incubated soil samples rich in organic carbon that stem from organic/peat soils or mineral soils rich in carbon. The results are related to possible, mostly chemical predictors to evaluate what drives decomposition of SOC in these soils. The topic is relevant and timely and I started reading with keen interest. However, the study has substantial methodological shortcomings which make it impossible to answer the aims listed at the end of the introduction. These are:

Soil samples were dried at 40 °C and then rewetted. This is a real shortcoming of the study for many reasons – destruction of part of the microbial biomass, shrinking and formation of hydrophobic surfaces in organic soils, stimulation of decomposition through rewetting in soils with smaller SOC contents etc. A sample from a Histosol

(samples from intact peatlands and other high SOM-soils are explicitly included in the study), dried at that temperature, can become a brick stone. It is unclear, how this influences the $CO_2$ release in different soils, particularly because wettability changes. Hence, how strongly drying effects SOC decomposability after rewetting depends on the sample's organic matter content! This is, in a negative sense, a good example that a uniform sample treatment does not guarantee the magnitude and direction of the analytical error to be the same. Further, how were intact, fibric peat samples passed through a 2 mm mesh?

The chosen incubation time is very short (40 – 90 hours). The authors speak about 'relatively constant basal respiration', but usually respiration rates decline exponentially over time (see also Fig. A1). What is measured during those 2-4 days is mostly the decomposition of fresh plant residues, which are abundant in the samples at different amounts. These amounts depend on sampling depth, land-use, and time of sampling, the latter also with reference to management activities (e.g., time passed since last cut of grass, position in the crop rotation etc.). It does not tell much about the decomposability of bulk SOC but this is, what the authors were interested in. For this purpose, incubations of many months are necessary in order to get a substantial contribution in $CO_2$ also from carbon not bound to fresh plant residues.

These factors might explain, why no correlations between soil properties and soil basal respiration could be identified and why "a decreasing trend of specific basal respiration with higher SOC was identified". To my opinion the latter finding clearly indicates, that high-C-samples suffered the most from drying, in accordance to the expected change in physical properties discussed above.

I am sorry, but these points are too important to let this manuscript pass. I therefore do not list further comments and suggestions.

---

## Referee Comment (RC3) · Anonymous Referee #3 · 19 Jun 2017

General Comments: The present article studies SOM depleted peatland soils, which where intensively used for agriculture and therefore lost much of their initial C. Soils were classified into different disturbance classes and were incubated to measure $CO_2$-emissions and to assess their sensitivity to mineralization. The research topic is very interesting and of great scientific significance as these soils become more and more relvant but the article should be written more clearly and concisely. The results and discussion section should focus more on the central aims of the study. Beside this, there are some fundamental methical problems and open questions which make it hard to interpret the obtained results. Specific comments: L 95: . . . samples of organic and mineral soil horizons. . . L 96 please give more information about the sampled horizons (exact sampling depth, time of sampling) L126 Was the time of sampling

prior to incubation? L 129 I suppose that 40° C is too high and produces artefacts, so that the composition of the microbial community might be strongly affected. Milder drying at room temperature or incubation of field-moist samples would have been more adequate. L 131 This is unclear. By stirring soil samples, their undisturbed pore distribution gets lost. Please describe in more detail what is meant by 60% water-filled pore space and how it is reached. L 139 40 to 90 h of incubation seem to short to really measure the sensitivity of old peat SOM rather than the mineralization of newly incorporated litter. L 172: The authors should generally rethink their classification system for the degree of disturbance (Table A1). The authors define disturbance according to pedogenic features and separate peats under permanently saturated conditions (Hr) from undrained surface peats (Hw). Both peat classes appear in peatlands with high water tables and do not show any pedogenic alterations caused by human impact. Furthermore, earthified horizons are described as moderately disturbed (D2), whereas aggregate and shrinkage horizons are classified as strongly disturbed (D3). This is not true, since peat in earthified horizons is highly decomposed (no visible plant residues') whereas shrinked or aggregated peat is particularly lesser decomposed. In addition, moorshified peats were classified as heavy disturbed as well as mixtures of peat and mineral soil. I would suggest to separate these two groups as there are fundamental differences in soil formation and C contents.

Results and discussion: Due to methical problems and questions stated above, it is difficult to interpret the obtained results, especially when regarding the disturbance of the peats or the quality of SOM. The authors should consider and discuss these methical problems. Nevertheless, there are some interesting results, e.g. showing that characteristics of bogs and fens become more and more similar after intensive drainage.

---

## Author Comment (AC1) · 24 Jul 2017

Responses to Referee #1

Referee #1 brought our attention to a problem with our statistical analysis and criticised the length and readability of the manuscript. We greatly appreciate pointing us to the issue of the unbalanced experimental design. The statistical soundness of the manuscript could therefore be considerably improved (see comment 1.5).

1.1) In general the article is too long, authors could consider to make the article concise and focused on key results. In the current form it is too hard to get bottom of the core results found.

We agree with the reviewer that the paper can be shorted and will refrain from showing

results of the original gls model as the results did not offer substantially stronger insight than the regression analysis but obviously led to some confusion (e.g. comment 1.4). Furthermore, we will delete the results on the microbial biomass and respiratory quotient to shorten the article as these data show very similar patterns as the respiration rates.

1.2) According to the authors, soil samples from Soil Inventory were used for the analysis, but did not mention the year of the sample collection. How old the samples were

The samples used in this study were collected between March 2011 and November 2014. As the selected samples were only a small percentage of the German Agricultural Soil Inventory the age of the stored samples varies accordingly. The sampling date will be added to the data table S1.

1.3) and was the van Post's disturbance class was identified during sample collection? Or did the authors assign the disturbances classes of the archive soils they used? It seems from Table A1 the disturbance classes need to be identified in field condition.

Yes, the decomposition after von Post and the soil horizons - which are even more important to assign disturbance classes - were determined in the field at a soil pit, which was dug at each site of the inventory. This will be clarified in the method's section. The disturbance classes were assigned by the authors based on all field and laboratory information available for the samples.

1.4) The description of results is confusing; for example "D0B samples had significantly lower (p < 0.01) SBR rates than the classes D4F," was it from the statistical analysis done with GLS model described in statistics section? Or from ANOVA and TukeyHSD?

All p-values given in the text had been calculated with an ANOVA and Tukey's test, except when referring to Spearman's correlation coefficients. However, we will replace the ANOVA by a gls model due to the unbalanced sample design. In any case, we will make sure to be clearer on which statistical analysis was used.

1.5) The experimental design was unbalanced, authors need to describe how they handle "unbalanced" issue.

We mistakenly used an ANOVA which cannot handle unbalanced data. To analyse the data on significant differences, e.g. between the different disturbance classes, the authors will now use a generalized least squares model (gls of "nlme" R package) followed by Tukey's test. By using the variance structure varIdent it is possible to handle unbalanced data by taking into account the specific variances of the different classification factors.

1.6) What are the explanatory variables involved in ANOVA.

We tested differences between the disturbance classes.

1.7) Another example (not limited to) of confusing description of results can be found in L231-236. Were those correlation coefficients significant?

Yes, the Spearman correlation coefficients were significant ($p < 0.05$) as stated in the beginning of L231 and L233. The section will be re-written to improve clarity.

1.8) Need to mention the step wise variable section using GLS to identify the variables that explained most variation.

This issue is no longer of relevance as the old gls model will be removed to clarify and shorten the manuscript.

1.9) Authors need to think about making their description and presentation less convoluted for the readers.

We are confident that the manuscript will become more concise and less convoluted when removing the old gls as well as the results on the microbial biomass and the respiratory quotient. Furthermore, we will carefully edit the manuscript for clarity.

1.10) L249-251 was it based on Spearman's rank correlation?

No, these results were based on ANOVA and Tukey's test. The section will be rewritten due to the change from ANOVA to gls.

1.11) L253-259 need to rewrite in clear concise manner.

We will rewrite this section in the course of the revision of the manuscript.

1.12) Figure 2 and 3: was there any difference in data presented? Or same data repeated.

The underlying incubation and soil property data is the same. Figure 2 gives an overview of significant ($p < 0.05$) correlations between specific basal respiration and soil properties. There is no grouping by disturbance class. Figure 3 shows the relationship between important soil properties and specific basal respiration rates (SBR), which might not be significant. While Figure 2 is, in our opinion, important as a summary of our data, Figure 3 shows more detailed information such as the high variability of the SBR at low SOC concentrations or the non-linear nature of the relationship between CN-ratio and SBR.

1.13) It would be better if the authors used the macromolecular quality not only CN ratio of organic matter to assess the effect of disturbance on SOC decomposition.

We are not sure what type of analysis or data the reviewer is suggesting here, but, for example, NMR might have offered more information on SOM quality. However, this study is unique due to its large range of sampling sites and the costs of such analyses of so many samples were not covered by project funding. Furthermore, some other methods such as density fractionation are not yet well tested for organic soils. Therefore, we "only" used C/N and isotopic analyses to describe the SOM quality beside the Post degradation state.

1.14) Equation in L149 doesn't make sense

The equation is correct. The temporal development of the respiration from initial to equilibrium rate is described by (a-SBR)(1-e-kt). It is a positive value with the unit of

the respiration rate. This value is subtracted from the initial respiration a, which results in CO2-C(t). We will make sure that formatting of CO2-C(t) is with an En dash and no spaces and not with an Em dash and spaces, this was mistakable and might have been understood by the reviewer as CO2 minus C(t) which would indeed not have made sense.

1.15) what does L143-144 refer to?

This sentence solely states that R was used for all statistical analyses in this manuscript.

Responses to Referee #2

The authors appreciate the comments of the referee concerning the length of our incubation and the pre-treatment of our samples. Overall, we agree that the results are difficult to interpret, but we don't agree that this is due to fundamental methodological issues, but due to the high variability of soil properties and soil forming processes in such "complicated" soils. However, we understand the reviewer's concerns and agree that these are issues which need to be discussed. Therefore, we will add a sub-section in the discussion section in which we will critically discuss the major methodical issues pointed out by the referees.

2.1) Soil samples were dried at 40°C and then rewetted. This is a real shortcoming of the study for many reasons

We can understand criticism about the drying of the samples. Before replying to the individual issues pointed out by the referee, we want to emphasize two points: 1) We agree that drying will affect e.g. the microbial community. However, these effects are not necessarily long-lasting. For example, Haney et al. (2004) have shown that respiration rates of field-moist samples from different mineral soils and those of samples dried at 40°C became indistinguishable after four days of incubation or even earlier. Even when samples have been dry for a year, microorganism react strongly different within

the first few hours after re-wetting, but respiration becomes much more similar within the first 5 days of incubation (Meisner et al., 2013). Similarly, drying and rewetting effects were minor for organic and paddy soils (Grover and Baldock, 2012; Wang et al., 2015). To minimise re-wetting effects, we used a – compared to other studies – relatively long pre-incubation period of two weeks. In addition, using dried and re-wetted samples in combination with pre-incubation is relatively common even for organic soils (Béasse et al., 2015; Taggart et al., 2012). 2) We would also like to emphasize that our sample set was derived from the German Agricultural Soil Inventory which sampled more than 3100 sites (mineral and organic soils), and that the time frame of sampling was nearly 4 years. From this inventory, we selected 91 samples based on a cluster analysis which resulted in a unique sample set that cover country scale variability. On the one hand, we agree that the use of fresh samples would be preferable for incubation experiments, but on the other hand, huge sample sets covering many sites over large geographical extents can realistically only be collected during very large projects, and such projects have their logistical constraints such as long-term storage capacity for frozen or field-moist samples.

2.2) destruction of part of the microbial biomass

As drying affects the microbial biomass, we pre-incubated the rewetted samples aerobically for 14 days. For example, Scheu and Parkinson (1994) have shown that drying only caused a slight reduction (< 10 %) in microbial biomass, and Wang et al. (2015) found no storage effects on microbial biomass for paddy soils.

2.3) shrinking and formation of hydrophobic surfaces in organic soils

Hydrophobicity is indeed well known for organic soils, but is generally strongly dependent on the soil moisture and often reversible (Doerr et al., 2000). In any case, we rarely observed hydrophobicity when rewetting the samples and this was overcome by both applying dispersed water and gentle stirring of the samples for uniform moistening and by pre-incubation.

2.4) stimulation of decomposition through rewetting in soils with smaller SOC contents etc.

We do not understand why rewetting should only stimulate decomposition in soils with smaller SOC contents, but would assume that this would affect high-C-samples as well. To our knowledge, there is no literature comparing the effects of rewetting on soils with different SOC contents.

2.5) A sample from a Histosol (samples from intact peatlands and other high SOM-soils are explicitly included in the study), dried at that temperature, can become a brick stone. It is unclear, how this influences the CO2 release in different soils, particularly because wettability changes. Hence, how strongly drying effects SOC decomposability after rewetting depends on the sample's organic matter content! This is, in a negative sense, a good example that a uniform sample treatment does not guarantee the magnitude and direction of the analytical error to be the same.

Yes, indeed. But drying a clay sample also results in a brick stone. Wettability effects surely play a large role during the process of rewetting (and even more so in the field where drying and rewetting alternate), but due to the process of slowly adding water and the pre-incubation we did not monitor the CO2 production during this phase. Furthermore, hydrophobicity is not restricted to peat, but a very frequent feature also of mineral soil (e.g. Doerr et al., 2006). Overall, the interaction between wettability and decomposition is a highly interesting topic but beyond the scope of our study. We do not agree using different pre-treatments would make sense as we do not have "high C" and "low C" samples – where would be the threshold in such a continuum? – and as there is no clear indication from literature on "how strongly drying effects SOC decomposability after rewetting depends on the sample's organic matter content".

2.6) Further, how were intact, fibric peat samples passed through a 2 mm mesh?

The peat was grated using the mesh or cut to 2 mm pieces (e.g. Eriophorum peat) with scissors.

2.7) The chosen incubation time is very short (40 – 90 hours).

The incubation time of this study was based on the standard procedure for this kind of incubation experiment which is 24 hours or less (e.g. Anderson and Domsch, 1986, 1993; Anderson and Joergensen, 1997; Blagodatskaya et al., 2014; Blagodatskaya and Anderson, 1998; Böhme et al., 2005; Dilly and Munch, 1998, 1996; Kautz et al., 2004; Müller and Höper, 2004). As pre-experiments have shown that this is too short for some organic soil samples to reach quasi-steady state respiration rates we increased the incubation time up to 90 h. Furthermore, we kept the incubation time as short as possible to prevent microorganisms from "starving out" which might happen in long incubation experiments (Holland et al., 1995), where there are, in contrast to natural environments, no processes that provide new supplies of organic matter and nutrients for the microbial biomass (e.g. drying and rewetting, fresh root exudates). Therefore, one might even argue that an incubation study becomes even less close to nature the longer the duration of incubation is.

2.8) The authors speak about 'relatively constant basal respiration', but usually respiration rates decline exponentially over time (see also Fig. A1).

Obviously we did not explain our methods clearly enough. Of course the respiration rates decline exponentially over time as shown in Fig. A1, and thus we fit an exponential model to our all three triplicates of each sample (equation 1). The asymptotic value of the model is the value of "basal respiration" which we used for further analysis. Furthermore, we have quantified the uncertainties of the basal respiration using the DREAM-algorithm (L 152ff). As shown in Fig. 4a, most of the uncertainties are low or even too small to be visible in this graph (values are also given in the data table S1). Applying this method, we are able to objectively quantify the respiration rates and their uncertainties. We would also like to emphasize that our method is transparent in the calculation of the SBR and that the estimation of uncertainties is beyond of what most previous studies did.

2.9) What is measured during those 2-4 days is mostly the decomposition of fresh plant residues, which are abundant in the samples at different amounts. These amounts depend on sampling depth, land-use, and time of sampling, the latter also with reference to management activities (e.g., time passed since last cut of grass, position in the crop rotation etc.). It does not tell much about the decomposability of bulk SOC but this is, what the authors were interested in. For this purpose, incubations of many months are necessary in order to get a substantial contribution in $CO_2$ also from carbon not bound to fresh plant residues.

Please refer to comment 2.7 regarding the incubation time. To reduce the potential problem of fresh plant residues in the samples, we did not use samples from 0-10 cm in our experiment, which contain most roots in grasslands. Furthermore, we excluded sites with recent fertilisation before sampling. Roots have been separated by hand, and as this is generally easier for soils with low organic matter content, potentially remaining roots might, in our opinion, not be the reason for higher respiration rates of samples with a lower SOM content. Finally, when including land-use, sampling month and depth in the (original) gls model none of the three variables were significant.

2.10) These factors might explain, why no correlations between soil properties and soil basal respiration could be identified and why "a decreasing trend of specific basal respiration with higher SOC was identified".

A recent incubation study on disturbed fen soils by Bader et al. (2017) incubated non-dried samples for over 6 months. Yet, they also could only find weak relationships between $CO_2$ emissions and soil chemical and physical properties (SOC content, bulk density and CN ratio). These results underline that explaining respiration rates from disturbed organic soils is complex, and that a high variability remains even when the incubation methods are more in line with the reviewers' suggestions. Furthermore, Bader et al. (2017) provide a literature review on $CO_2$ emissions from incubation experiments on mineral and peat samples. The range of the specific respiration rates measured in our study (0.03 to 0.6 mg $CO_2$-C g $SOC^{-1}$ $d^{-1}$) agree with the range of values that can

be expected at 23 °C, i.e. there is no indication for a systematic error due to our sample preparation and measurement protocol (Figure 4 in Bader et al., 2017). In addition, due to the strong degradation of many of our drained sites, the topsoils tend to have lower SOC contents than the subsoils. Therefore, many of the samples with high SOC contents are subsoil samples. Several studies have shown that the availability of labile compounds decreases with depth due to an older age (e.g. Scanlon and Moore, 2000 or Wang et al., 2010; summarized in Bader et al., 2017). Therefore, in our case, lower respiration rates for high SOC samples are plausible in our opinion. This aspect and the according references will be added to a revised manuscript.

2.11) To my opinion the latter finding clearly indicates, that high-C-samples suffered the most from drying, in accordance to the expected change in physical properties discussed above.

From our point of view, our data set is not suitable to prove or disprove the opinion of the reviewer, which we do not share.

References

Anderson, T.-H. and Domsch, K. H.: Carbon assimilation and microbial activity in soil, Zeitschrift für Pflanzenernährung und Bodenkd., 149(4), 457–468, 1986.

Anderson, T.-H. and Domsch, K. H.: The metabolic quotient for CO2 (qCO2) as a specific activity parameter to assess the effects of environmental conditions, such as ph, on the microbial biomass of forest soils, Soil Biol. Biochem., 25(3), 393–395, 1993.

Anderson, T.-H. and Joergensen, R. G.: Relationship between SIR and FE estimates of microbial biomass C in deciduous forest soils at different pH, Soil Biol. Biochem., 29(7), 1033–1042, 1997.

Bader, C., Müller, M., Schulin, R., and Leifeld, J: Peat decomposability in managed organic soils in relation to land-use, organic matter composition and temperature, Biogeosciences Discuss., https://doi.org/10.5194/bg-2017-187, 2017.

Béasse, M. L., Quideau, S. A. and Oh, S.-W.: Soil microbial communities identify organic amendments for use during oil sands reclamation, Ecol. Eng., 75, 199-207, 2015.

Blagodatskaya, E., Blagodatsky, S., Anderson, T. H. and Kuzyakov, Y.: Microbial growth and carbon use efficiency in the rhizosphere and root-free soil, PLoS One, 9(4), 2014.

Blagodatskaya, E. V. and Anderson, T. H.: Interactive effects of pH and substrate quality on the fungal-to-bacterial ratio and QCO2 of microbial communities in forest soils, Soil Biol. Biochem., 30(10–11), 1269–1274, 1998.

Böhme, L., Langer, U. and Böhme, F.: Microbial biomass, enzyme activities and microbial community structure in two European long-term field experiments, Agric. Ecosyst. Environ., 109, 2005.

Dilly, O. and Munch, J.-C.: Ratios between estimates of microbial biomass content and microbial activity in soils, Biol. Fertil. Soils, 27, 374–379, 1998.

Dilly, O. and Munch, J. C.: Microbial biomass content, basal respiration and enzyme activities during the course of decomposition of leaf litter in a black alder (Alnus glutinosa (L.) Gaertn.) forest, Soil Biol. Biochem., 28, 1073–1081, 1996.

Doerr, S.H., Shakesby, R.A., Dekker, L.W., and Ritsema, C. J.: Occurrence, prediction and hydrological effects of water repellency amongst major soil and land-use types in a humid temperate climate, Europ. J. Soil Sci., 57, 741-754, 2006.

Grover, S. P. P. and Baldock, J. A.: Carbon chemistry and mineralization of peat soils from the Australian Alps, Eur. J. Soil Sci., 63(2), 129–140, 2012.

Haney, R. L., Franzluebbers, A. J., Porter, E. B., Hons, F. M. and Zuberer, D. A.: Soil carbon and nitrogen mineralization: influence of drying temperature, Soil Sci. Soc. Am. J., 68, 489–492, 2004.

Holland, E. A., Townsend, A. R. and Vitousek, P. M.: Variability in temperature regulation of CO2 fluxes and N mineralization from five Hawaiian soils: implications for a changing climate, Glob. Chang. Biol., 1, 115–123, 1995.

Kautz, T., Wirth, S. and Ellmer, F.: Microbial activity in a sandy arable soil is governed by the fertilization regime, Eur. J. Soil Biol., 40(2), 87–94, 2004.

Meisner, A., Bååth, E. and Rousk, J.: Microbial growth responses upon rewetting soil dried for four days or one year, Soil Biol. Biochem., 66, 188-192, 2013.

Müller, T. and Höper, H.: Soil organic matter turnover as a function of the soil clay content: Consequences for model applications, Soil Biol. Biochem., 36(6), 877–888, 2004.

Scanlon, D., and Moore, T.: Carbon dioxide production from peatland soil profiles: the influence of temperature, oxic/anoxic conditions and substrate, Soil Sci., 165, 153–160. 2000.

Scheu, S. and Parkinson, D.: Changes in bacterial and fungal biomass C, bacterial and fungal biovolume and ergosterol content after drying, remoistening and incubation of different layers of cool temperate forest soils, Soil Biol. Biochem., 26, 1515-1525, 1994.

Taggart, M., Heitman, J. L, Shi, W. and Vepraskas, M.: Temperature and water content effects on carbon mineralization for sapric soil material, Wetlands, 32, 939–944, 2012.

Wang, X., Li, X., Hu, Y., Lv, J., Sun, J., Li, Z. and Wu, Z.: Effect of temperature and moisture on soil organic carbon mineralization of predominantly permafrost peatland in the Great Hing'an Mountains, Northeastern China, J. Environ. Sci., 22, 1057–1066, 2010.

Wang, J., Chapman, S. J., and Yao, H: The effect of storage on microbial activity and bacterial community structure of drained and flooded paddy soil, J. Soils Sediments, 15, 880–889, 2015.

Responses to Referee #3

We greatly appreciate the comments of the referee which helped us to improve the manuscript especially in terms of the ambiguities and uncertainties regarding the methods. 3.1) L 95:...samples of organic and mineral soil horizons

We will change the text according to the reviewer's suggestion.

3.2) L 96 please give more information about the sampled horizons (exact sampling depth, time of sampling)

The depth increment of each sample can be found in the supplementary data file S1. The samples used in this study were collected between March 2011 and November 2014; and the sampling data will be added to the data table S1. A reference to the data table S1 will be made in this section to clarify where to find this information.

3.3) L126 Was the time of sampling prior to incubation?

We suppose the reviewer asks for the time between sampling and incubation here. Yes, the samples were taken up to 4 years prior to incubation as they are part of a very large project (see also comment 2.1).

3.4) L 129 I suppose that 40âŮęC is too high and produces artefacts, so that the composition of the microbial community might be strongly affected. Milder drying at room temperature or incubation of field-moist samples would have been more adequate.

Regarding the issue of using dried samples, please refer to the response to comments 2.1 and 2.2. In addition, we doubt that drying at room temperature would be a promising alternative: For peat samples, this process might take weeks (if not months) during which microorganism would have ample opportunity to decompose and alter the SOM.

3.5) L 131 This is unclear. By stirring soil samples, their undisturbed pore distribution gets lost.

For uniform rewetting, the calculated amount of water (for details, see answer to next

comment) was applied using a squirt bottle under continuous gentle stirring of the samples with a spoon. The undisturbed pore distribution has already been destroyed by sieving the samples to 2 mm. Incubation experiment using disturbed samples is common practice, even for peat soils (e.g. Brake et al., 1999; Brouns et al., 2016; Urbanová et al., 2011) and often caused by logistical constraints (see comments 2.1 and 2.2).

3.6) Please describe in more detail what is meant by 60% water-filled pore space and how it is reached.

The dried (40 °C) and < 2 mm sieved soil samples were moistened to a standardised water content of 60 % water-filled pore space. To do so, the apparent porosity of the dried and sieved sample was calculated as follows: $\Phi$=1- _loose/_s , where loose [g cm-3] is the bulk density of the loose sample and s [g cm-3] is the soil particle density. The soil particle density s was determined after Bohne (2005): _s=0.086*AC+1.44, where AC [%] is the ash content of the sample. The necessary amount of water to reach 60 % water-filled pore space was then applied to the soil samples under continuous stirring with a spoon to ensure uniform rewetting.

3.7) L 139 40 to 90 h of incubation seem to short to really measure the sensitivity of old peat SOM rather than the mineralization of newly incorporated litter.

Please see our response to comment 2.7 and 2.9.

3.8) L 172: The authors should generally rethink their classification system for the degree of disturbance (Table A1). The authors define disturbance according to pedogenic features and separate peats under permanently saturated conditions (Hr) from undrained surface peats (Hw).

While the reviewers are correct that an Hr horizon is one under permanently saturated conditions, Hw horizons are characterised by "temporary water saturation within the amplitude of the groundwater, with signs of oxidation" (Ad-Hoc-Arbeitsgruppe Boden,

2005) and thus by temporarily unsaturated conditions. However, this is not restricted to topsoils of (semi-natural) peatlands. As in our sample set, Hw horizons frequently are subsoil horizons of drained peatlands. In accordance with the definition, we are convinced these have been influenced by aerobic conditions and have thus to be differentiated from permanently water-saturated horizons. Although secondary pedogenic features might not be visible yet, the SOM has probably already been altered to a certain extent. Generally, we are aware of the fact that there are numerous soil classification systems, and that the German one might not be perfect regarding organic soils. Nonetheless, the German classification system has been used in the German Agricultural Soil Inventory as this is the common taxonomy for German soil maps. For sure, we cannot expect readers to be familiar with this specific soil taxonomy and will therefore amend the explanations on the horizons.

3.9) Furthermore, earthified horizons are described as moderately disturbed (D2), whereas aggregate and shrinkage horizons are classified as strongly disturbed (D3). This is not true, since peat in earthified horizons is highly decomposed (no visible plant residues') whereas shrinked or aggregated peat is particularly lesser decomposed.

As explained above, our classification system is in accordance with the German soil taxonomy, which clearly defines earthified horizons (Hv) as "topsoil horizons of moderately drained or extensively used peatlands" and aggregate (Ha) and shrinkage (Ht) horizons as "subsoil horizons of heavily drained peatlands" and "peat shrinkage horizon", respectively (Ad-Hoc-Arbeitsgruppe Boden, 2005). However, the reviewer is correct in the assessment that plant residues might still be visible in aggregate or shrinkage horizons. On the other hand, the pedogenic transformation of a drained peatland starts with the formation of an earthified topsoil horizon. If the drainage is continued or intensified, the subsoil peat might start to develop secondary pedogenetic features (aggregates, cracks). Therefore, the occurrence of such features points to intensive anthropogenic disturbance, and there would be no Ha or Ht horizons without a prior formation of an Hv horizon. In the end, this comes down to the question whether we

base our classification on the degree of decomposition (which is anyway difficult for drained soils as explained in L338 ff.) or on the soil taxonomy. However, as we did use the disturbance classes as categorical, but not as numerical variables in our statistical analyses, this question might even be not that important as long as the classification criteria are transparent.

3.10) In addition, moorshified peats were classified as heavy disturbed as well as mixtures of peat and mineral soil. I would suggest to separate these two groups as there are fundamental differences in soil formation and C contents.

We tried to separate this disturbance class, but – despite of farmer's information on the land-use history of sites – in a lot of cases it remained unclear whether the peat is "purely" moorshy or (naturally or anthropogenically) mixed with mineral soil. In theory, there might be clear differences in the C content as postulated by the reviewer, but as shown in Figure 3a, there is rather a continuum of C contents. Furthermore, there are also cases were moorshified peat has been mixed with sand, which complicates the situation even further. In this context, we would like to emphasise that the sampling sites were not picked with the aim of representing clear-cut types of organic soils, but randomly by an 8x8 km grid according to the protocol of the German Agricultural Soil Inventory. This might be seen as a weakness, but in fact, the complicated nature of many soil profiles shows both how underappreciated and how frequent such complex organic soils are as they are understandably not sampled when looking for well-defined "nice" soil profiles.

3.11) Results and discussion: Due to methical problems and questions stated above, it is difficult to interpret the obtained results, especially when regarding the disturbance of the peats or the quality of SOM. The authors should consider and discuss these methical problems.

We agree that the results are difficult to interpret, but we don't agree that this is due to fundamental methodological issues, but due to the high variability of soil properties

and soil forming processes in such soils. However, we understand the concerns and will add a sub-section in the discussion section in which we will critically discuss the major methical uncertainties pointed out by the referees.

References

Ad-Hoc-Arbeitsgruppe Boden: Bodenkundliche Kartieranleitung KA5 (Manual of soil mapping), 5th ed., E. Schweizerbart'sche Verlagsbuchhandlung, Hanover, Germany, 2005.

Bohne, K: An introduction into applied soil hydrology. Schweizerbart'sche Verlagsbuchhandlung, Stuttgart, 2005.

Brake, M., Höper, H. and Joergensen, R. G.: Land use-induced changes in activity and biomass of microorganisms in raised bog peats at different depths, Soil Biol. Biochem., 31(11), 1489–1497, 1999.

Brouns, K., Keuskamp, J. A., Potkamp, G., Verhoeven, J. T. A. and Hefting, M. M.: Peat origin and land use effects on 535 microbial activity, respiration dynamics and exo-enzyme activities in drained peat soils in the Netherlands, Soil Biol. Biochem., 95, 144–155, 2016.

Urbanová, Z., Picek, T. and Bárta, J.: Effect of peat re-wetting on carbon and nutrient fluxes, greenhouse gas production and diversity of methanogenic archaeal community, Ecol. Eng., 37, 1017-1028, 2011.